# Comparison of Differentially Expressed Genes in Human and Canine Osteosarcoma

**DOI:** 10.3390/life15060951

**Published:** 2025-06-12

**Authors:** Jorja Jackson-Oxley, Aziza A. Alibhai, Jack Guerin, Rachel Thompson, Rodhan Patke, Anna E. Harris, Corinne L. Woodcock, Dhruvika Varun, Maria Haque, Tinyiko K. Modikoane, Amber A. Kumari, Jennifer Lothion-Roy, Simone de Brot, Mark D. Dunning, Jennie N. Jeyapalan, Nigel P. Mongan, Catrin S. Rutland

**Affiliations:** 1School of Veterinary Medicine and Science, Faculty of Medicine and Health Sciences, University of Nottingham, Nottingham LE12 5RD, UK; svxjj1@exmail.nottingham.ac.uk (J.J.-O.); svaaa3@exmail.nottingham.ac.uk (A.A.A.); jackguerin@btinternet.com (J.G.); styrt5@exmail.nottingham.ac.uk (R.T.); stxrp16@exmail.nottingham.ac.uk (R.P.); svzaeh@exmail.nottingham.ac.uk (A.E.H.); stxdv8@exmail.nottingham.ac.uk (D.V.); stxmh36@exmail.nottingham.ac.uk (M.H.); stxtm23@exmail.nottingham.ac.uk (T.K.M.); mzyak32@exmail.nottingham.ac.uk (A.A.K.); svzjhl@exmail.nottingham.ac.uk (J.L.-R.); simone.debrot@unibe.ch (S.d.B.); svzmdd@exmail.nottingham.ac.uk (M.D.D.); plzjnj@exmail.nottingham.ac.uk (J.N.J.); catrin.rutland@nottingham.ac.uk (C.S.R.); 2Biodiscovery Institute, Faculty of Medicine and Health Science, University of Nottingham, Nottingham NG7 2RD, UK; nigel.mongan@nottingham.ac.uk (N.P.M.); 3COMPATH, Institute of Animal Pathology, University of Bern, 3012 Bern, Switzerland; 4Willows Veterinary Centre and Referral Service, Solihull B90 4NH, UK; 5Department of Pharmacology, Weill Cornell Medicine, New York, NY 10065, USA

**Keywords:** ASPN, BAMBI, bone cancer, diagnostics, STK3, therapeutics, osteosarcoma

## Abstract

Osteosarcoma (OSA) is the most prevalent bone malignancy in people and dogs. Current survival rates show the need for advances in novel therapies to help overcome the growth, survival and metastatic progression of the cancer. Canine models are often used to advance prognostic and treatment opportunities for OSA due to the similarities in the disease between species. This study focusses on the genetic and molecular similarities of OSA between human and canine specimens. Differentially expressed genes (DEGs) were compared and identified in canine and human OSA tumours, revealing 86 common genes, 36 having high and 50 having low expression. Further immunohistochemical analysis of the corresponding proteins of three identified DEGs (ASPN, STK3, BAMBI) allowed for the visualisation of protein expression in canine OSA tissues (n = 19). Overall nuclear and cytoplasmic H-scores were generated, and nuclear and cytoplasmic scores in males and females and in different anatomical locations (axial versus appendicular) were also investigated, presenting unique opportunities to understand the expression in this cancer type. This study contributes to a deeper knowledge of genetic pathways changes and identifies avenues for the diagnosis, prognosis and treatment of OSA in people and dogs, whilst encompassing the One Health concept in medicine.

## 1. Introduction

Osteosarcoma (OSA) is a highly metastatic bone malignancy usually affecting children and young adults, with an incidence rate of approximately 1.02/100,000 of the population (across all ages) [1,2,3]. These tumours are mesenchymal in origin and comprise of osteoid-forming cells, allowing lesions to have heterogenous histological manifestations [4]. OSA can be subcategorized depending on the major stroma present, for example, osteoblastic, chondroblastic, fibroblastic or small-cell [5]. To date, little research has been conducted in relation to the prognostic significance of OSA subclassifications or cellular components, in people or dogs [6].

OSA in people has many similarities, both clinically and molecularly, to the disease in domestic dogs (*Canis familiaris*), including lesion location and metastatic progression [1,7,8]. Canine OSA has a higher prevalence than that in people, estimated at approximately 13.9/100,000 [2]. It has therefore been suggested that spontaneous OSA in dogs can serve as a model system for translational cancer therapeutics in people, due to their many shared attributes [9]. Unveiling the molecular mechanisms behind the neoplasm in canines, and using knowledge from prior research conducted on human OSA, can advance diagnostic, prognostic and treatment opportunities available for OSA in both species.

There are many risk factors predisposing people and dogs to OSA lesion formation, such as the height and mass of an individual [10]. Large and giant breed dogs (25–45 kg and >45 kg, respectively) are most susceptible to OSA development, with the highest annual occurrence seen in Scottish Deerhounds, Leonbergers, Great Danes, and Rottweilers (3.28%, 1.48%, 0.87%, and 0.84%, respectively) [11]. Dogs with chondrodystrophy were shown to be less prone to OSA compared to non-chondrodystrophic breeds (odds ratio (OR) of 0.13, 95% confidence interval of 0.11–0.16) [12]. Likewise, people who are classified as very tall (≥90th percentile), those who had a higher-than-average birth weight (≥4046 g), and individuals who are taller than average (51st–89th percentile) have increased risks of developing OSA (OR 2.6, 1.35 and 1.35, respectively) [13]. Notably, OSA has also been shown to be associated with enhanced carcinogenic susceptibility during bone growth, distinguishable by the presence of the lesions within the metaphyseal regions of weight-bearing bones, such as the distal femur and proximal humerus in both canines and people [7,14]. This is also linked to the peak incidence in humans presenting in adolescents and young adults under 24 years old [15,16]. Both species display a bimodal age distribution, but dogs have their peak incidence in older adults, with dogs over 7 years old accounting for 80% of cases [10]. In both species, the prevalence of OSA is positively skewed towards the male population and correlates with the average age at which each gender population attains their final adolescent height [9,16]. Although less applicable to humans, neutering dogs contributes towards an increased risk of lesion formation compared to those that remain entire, possibly due to decreased progesterone levels [7,17].

The prognosis of OSA is poor due to the disease’s high malignant propensity [18]. Approximately 80% of dogs die due to metastasis and patients have a median survival time of no greater than 5 months if treated by amputation or lesion excision alone [19,20]. In people, the 5-year survival rates have improved drastically since the 20th century due to the evolution of treatment and available therapies, from approximately 20% to >65% [21,22]. However, the high tendency of OSA cellular dissemination means 10–20% of diagnosed patients already have secondary lesion formation within the lungs (85%), bone (8–10%), and, occasionally, the lymph nodes [18,23]. Subclinical or micro-metastases can affect 80–90% of patients and are undetectable by current diagnostic methods [23].

The pathogenesis of OSA has been difficult to distinguish due to the many potential cellular origins, an absence of recognisable precursor lesions, and the genetic complexity of tumours [24]. The clinical presentation of the disease in canines is predominantly variable degrees of lameness, often associated with a bony or soft tissue mass [1,25]. Radiographs can be performed to facilitate a diagnoses and a definitive diagnosis, and the grade of the tumour, can be achieved by obtaining a biopsy [26]. Surgery (limb sparing or amputation) with adjunct chemo/immunotherapy is deemed the standard procedure for both canine and human OSA [19,27]. Limb-salvage surgery is favourable in certain sites over amputation to retain limb function in canines, but amputation remains the mainstay for surgical treatment of canine OSA [28,29]. In people, 90% undergo limb salvage surgery over amputation [30,31,32], whilst radiation can also be conducted to aid the removal of microscopic residual disease [33]. Combined drug therapies are more effective and are favoured over monotherapies as a higher number of pathways can be targeted [34]. In humans, high-dose methotrexate, doxorubicin and cisplatin (MAP) is an effective, well-known therapy used alongside limb-sparing surgeries providing 5-year survival rates of approximately 70% for patients with non-metastatic disease [35,36], decreasing to 10–40% in patients with metastases [35]. Ifosfamide can be used concomitantly to increase efficacy, providing a dose dependent response rate of 10–40% [36,37,38]. In canines, cisplatin or carboplatin have been reported as stand-alone therapies or in combination with doxorubicin [7]. Cisplatin provides a median survival rate of between 9 and 11 months when administered post amputation, with approximately 20% of patients surviving 2 years [39].

There is a wide range of genetic abnormalities associated with OSA including overexpression of certain oncogenes and transcription factors, and disruption to the normal functioning of tumour suppressor genes [40]. Notably, many of these defects are also present in many prevalent cancers and are not OSA-specific [24]. Common chromosomal features such as aneuploidy (both species have a 75% aneuploid DNA index) and genomic instability are also present in both canine and human OSA [41].

Given the low survival rates, poor responses to treatments and lack of developments relating to OSA diagnostics and therapeutics within the past four decades, new therapeutic targets and regimens for OSA are urgently required to improve the life expectancy of patients. Although there are many common features of the disease in humans and canines, a deeper understanding of the genetics and molecular mechanisms of the disease is useful in understanding the role of dogs in translational cancer therapeutics, and is also in itself a powerful tool for identifying diagnostic methods, prognostic factors and treatment pathways [42]. This study compared significant differential gene expression between canines and humans using previous RNA sequencing analysis by Simpson et al. (2020) and Yang et al. (2014) [43,44]. The overarching hypothesis was that canines and people would share common differentially expressed genes (DEGs) and pathways. Our results identified common DEGs and pathways in OSA and provided insights into the mechanisms contributing to the development and progression of disease in these species, and into potential treatment pathways. IHC and H-scoring are widely used diagnostic and prognostic techniques in human medicine [45,46]; yet, veterinary research into differentiated genes and identifying markers for IHC and H-scoring is in the developmental stage. Using these techniques, the second aim was to further investigate protein expression of three genes significantly overexpressed in both canine and human OSAs, namely *ASPN*, *STK3* and *BAMBI*. The hypothesis was that these proteins would be expressed in canine OSA cells (nuclear and cytoplasmic) in both sexes, and within differing OSA anatomical locations.

## 2. Materials and Methods

### 2.1. Genetic Analysis

#### 2.1.1. Differentially Expressed Gene Analysis

DEGs in both human and canine OSA tumours compared to patient-matched non-malignant tissues were compared to investigate genetic similarities between the two species. The human and canine RNA sequencing datasets, for both over- and underexpressed genes, were obtained from previously published studies by Yang and coauthors and by Simpson and colleagues [43,44]. The canine samples were excised from treatment-naïve patients (following amputation surgeries) [43]. The human samples were analysed as they represented a combination of tissues excised from bone tissues and cell lines, as well as metastatic disease. Whilst most of the human tumours/cell lines were treatment-naïve, full details were not published for every sample (GEO IDs: GSE14359, GSE16102, GSE12865, GSE11414, GSE42352, GSE36001, GSE32964 and GSE30807) [44]. Ensembl BioMart 113 was used to identify human orthologs of canine differentially expressed genes. Venny 2.1.0 was used to identify common DEGs between the two species [47]. RNA sequencing analysis and consequent DEG analysis were also conducted using the results from Poudel et al. (2024) and compared to the canine data using Venny 2.1.0 [48]. The results from the first comparison constituted the main focus of this study, as the human dataset was a meta-analysis of eight different GEO datasets, providing more comprehensive oversight compared to using any one individual cohort dataset. However, we acknowledge the fact there are other studies not included in this dataset, such as PMID: 29050494, which may prove useful for future analyses [49].

#### 2.1.2. STRING and KEGG Pathway Analysis

Over-representation KEGG pathway analysis using Webgestalt 2019 was performed to identify pathways related to the common over- and underexpressed DEGs, as identified in Section 2.1.1 [50]. The organism of interest when performing these searches was set to *homo sapiens*, although secondary checks confirmed there were no differences in results retrieved when the parameters were set to compare the data to *canis lupus familiaris*. Over-representation KEGG pathway analysis was also performed on significant over- and underexpressed genes for humans and canines individually. This allowed for comparison of the pathways potentially associated with OSA development and/or progression between the species. KEGG analysis also showed the involvement of two of the genes selected to immunohistochemical protein analysis in this present study (*STK3*, Hippo signalling pathway and *BAMBI*, TGF-β signalling). Hence, schematic diagrams were generated to show their involvement in these pathways (Figure A2). STRING analysis was also conducted to investigate protein–protein interactions in *Canis lupus familiaris* for the extracellular protein ASPN due to its absence in the KEGG analysis.

### 2.2. Immunohistochemistry

Three genes from those identified as commonly overexpressed in human and canine OSA were selected for protein expression investigation. IHC on canine OSA tumour samples was performed to identify cytoplasmic and/or nuclear protein expression encoded by these genes. Treatment-naïve canine samples from amputation surgeries or euthanized patients were utilised for IHC in this present study due to their availability. RNASeq data was also validated using qRT-PCR to confirm overexpression of all three genes. Gene expression confirmation of *ASPN* was previously conducted via qRT-PCR (TaqMan assay Cf02659289_m1, ThermoFisher Scientific, Waltham, MA, USA) [43]. RT-qPCRs were additionally conducted to confirm *STK3* and *BAMBI* gene expression (TaqMan assays Cf00355531_m1 and Cf02707430_s1, respectively; ThermoFisher Scientific, Waltham, MA, USA). Expression was normalised to the internal control, actin (TaqMan assay Cf04931159_m1, ThermoFisher Scientific, Waltham, MA, USA). Statistical *t*-tests were performed using GraphPad Prism (GraphPad Prism, version 9.4.0, Boston, MA, USA) to compare the relative expression of *STK3* and *BAMBI* in canine tumours compared to patient-matched non-malignant counterparts.

#### 2.2.1. Specimen Preparation and Ethics

Canine tissue samples were approved by the University of Nottingham School of Veterinary Medicine and Science ethics committee (ethics permission number 3832 2305020) and complied with international and national ethical standards. OSA samples were excised from Rottweilers euthanized or undergoing amputation surgery for reasons unrelated to this present study, with a definitive diagnosis of OSA given by a board certified histopathologist.

All Rottweiler OSA samples (n = 19) were obtained as formalin-fixed, paraffin-embedded (FFPE) tissue blocks. There were 12 females, 6 males and 1 with gender not specified. The age range of the females was between 4.83 and 12 years (mean 9.17 ± 0.689), and of the 12 females, 9 were neutered, 1 remained entire and 2 did not have their neutering status specified. Males had an age range between 4.5 and 9 years (mean 6.71 ± 0.655); within the six males, two were neutered, one remained entire and three did not specify the neutering status of the dogs. The females were statistically older than males at the time of neoplasm excision (*t*-test, *p* = 0.0341; Figure A1A); therefore, age was not analysed, but many samples were age-matched within the cohort. OSA samples were collected from the axial skeleton (n = 6) and appendicular locations (n = 13).

#### 2.2.2. Antibody Selection

*ASPN*, *STK3* and *BAMBI* were identified as being common overexpressed genes in both canine and human OSA tissue samples using the bioinformatic software previously stated in Section 2.1.1. Therefore, further investigations using IHC- and H-scoring techniques were conducted to show the cellular expression of these proteins. Antibodies used to detect the proteins encoded by these genes were selected using BLAST gene sequencing alignment software (v2.16.0). The antibodies for ASPN, STK3 and BAMBI were unconjugated rabbit polyclonal antibodies with proven IHC applications.

#### 2.2.3. Immunohistochemistry and Microscopy

Rottweiler OSA FFPE blocks were sectioned at 7 µm and IHC was performed using the Leica Novolink Polymer Detection Kit (Leica, Wetzlar, Germany) in accordance with the manufacturer’s protocol. Dilution optimizations resulted in 1:100 dilution for ASPN (HPA007894; Sigma-Aldrich, St. Louis, MO, USA), STK3 (12097-1-AP; ProteinTech, Rosemont, IL, USA), and BAMBI (PA5-38027; Thermo Fisher, Waltham, MA, UK). Primary antibodies were diluted in 5% foetal calf serum and negative controls were incubated using only foetal calf serum and no primary antibody.

Microscopy (Leica, Wetzlar, Germany) was conducted using systematic random sampling at 400× magnification, n = 5 photomicrographs per specimen, to allow for semi-quantitative analysis of cytoplasmic and nuclear protein expression within canine OSA tissue samples. Photomicrographs were also taken at 50× magnification for qualitative descriptions of staining patterns and identification of structures.

#### 2.2.4. H-Scoring and Statistical Analysis

Immunohistochemical interpretation using H-scoring methods is deemed ‘gold standard’ in human medicine; however, it is not as routinely used within veterinary pathology yet. This method semi-quantifies protein expression levels based on the intensity of the staining present. Typically, H-scores are calculated as an overall value (nuclear plus cytoplasmic), but increasingly separate nuclear and cytoplasmic scores are calculated as differentiation may provide clinically relevant information. Scores of 0, 1+, 2+, or 3+ (absent, weak, moderate, strong staining), were given for nuclear and cytoplasmic staining independently, and a H-score (0–300) was calculated using the following formula: H-score = [1 × (% cells 1+) + 2 × (% cells 2+) + 3 × (% cells 3+)]. This scoring method was performed for each protein of interest (ASPN, STK3 and BAMBI, n = 19) by one researcher, and an additional researcher randomly scored 10% of the samples to ensure concordance. An intraclass correlation coefficient (ICC) >90% was achieved for all proteins, confirming interpretation consistency. The mean standard error of the mean (SEM), and range (minimum and maximum scores) were calculated. The predominant staining intensity for both nuclear and cytoplasmic subcellular locations was described following H-score classifications (absent, low, moderate and high; Table 1). The general staining distribution was described for each protein (diffuse, multifocal or focal; Table 2). Low/moderate/high classifications were calculated based off the highest range (either nuclear or cytoplasmic) for each individual antibody: ASPN = low (≤48), moderate (49–96), high (≥97); STK3 = low (≤34), moderate (35–68), high (≥69); BAMBI = low (≤43), moderate (44–86), high (≥87). Graphs were generated for ASPN, STK3 and BAMBI to demonstrate staining intensities and score distributions (the figures in Section 3.2). Statistical *t*-tests were performed using GraphPad Prism (GraphPad Prism, version 9.4.0, USA) to check for significant differences between nuclear and cytoplasmic H-scores, male vs. female nuclear and cytoplasmic H-scores and appendicular vs. axial nuclear and cytoplasmic H-scores. Qualitative data was also utilised to contextualise immunohistochemical staining patterns.

## 3. Results

### 3.1. Genetic Analysis

#### Comparison of Significant DEGs in Canine and Human OSA

Two studies were used to identify commonly expressed genes between humans and canines, which had previously been shown to be expressed higher or lower than non-malignant specimens [43,44]. This study identified 36 overexpressed and 50 underexpressed genes common between the two species using Venny 2.1.0 (Figure 1A,B, Table A1) [43,44,47]. Pathway analysis of the common DEGs (both under- and overexpressed) revealed no significant pathways (Figure 1C and Figure 1D, respectively). However, this did provide insights into possible mechanisms and genes of interest that may represent potential targets for future therapies.

In the canine RNA sequencing analysis (using the published data [43]), 442 genes were identified as being overexpressed and 839 genes were underexpressed compared to patient-matched, non-tumour tissues (Figure 1E). The human RNA sequencing analysis (using a published dataset [44]) revealed 472 overexpressed and 507 underexpressed genes compared to non-malignant counterparts (Figure 1F).

KEGG pathway analysis was also performed for significantly over- and underexpressed genes separately for each species individually to compare the outcomes of the searches (Figure 1). Results showed that there were differences between the two species and the pathways involved in OSA development. The canine overexpressed gene search revealed three significant pathways (FDR-corrected *p*-value < 0.05), namely axon guidance, Hippo signalling and pathways in cancer (cfa04360, cfa04390 and cfa05200, respectively). There were also similarities to the pathways identified in the common overexpressed gene search results (Figure 1C,G), such as the involvement of transforming growth factor-beta (TGF-β) signalling (hsa04350 and cfa04350), Hippo signalling (hsa04390 and cfa04390) and other pathways in cancer (hsa05200 and cfa05200). Conversely, the pathways involved in the human overexpressed gene analysis revealed an array of different significant pathways (FDR-corrected *p*-value < 0.05; Figure 1I), such as those involved with immunity and infection (hsa04612, hsa04672, hsa05171 and hsa05150) as well as cell cycle pathways, including those regulating DNA replication (hsa04110).

Likewise, there were many differences between the underexpressed gene analysis of the common DEGs in canines and people (Figure 1D, Figure 1H and Figure 1J, respectively). However, the main focus was to investigate the overexpression of genes and potential therapeutic targets that restore dysregulation, although the commonly underexpressed gene *NQO1* was also investigated further within this study (hsa05200).

Pathways in cancer appeared as a significantly overexpressed pathway in canines (cfa05200) but was a significantly underexpressed pathway in humans (hsa05200). Cyclin D2 (*CCND2*) had opposing expression levels at the RNA level between species within this pathway; hence, this gene was explored in this study. Although this was not significant, a GAP junction (hsa04540) appeared within the KEGG analysis search when investigating the common overexpressed genes in human and canine OSA. *GJA1* appeared as a common DEG within this pathway and was investigated further within this study. We previously investigated LEF1 expression in canine OSA, and have revisited it within the present study, revealing that there was also significant overexpression of *LEF1* in human OSA [51].

The human dataset studied in this research was a meta-analysis study which included a range of cohorts [44]. A secondary differential gene analysis was additionally conducted using the mRNA sequencing data available from 89 of the 90 GEO datasets used in the Poudel et al. (2024) whole-transcriptome analysis (data was unavailable for one sample) [48]. The DEGs identified in the canine and both human datasets revealed that 33/36 of the common overexpressed and 38/50 of the common underexpressed DEGs were significant between all three analyses [43,44,48]. The three genes chosen for immunohistochemical analysis in this study (*ASPN*, *STK3* and *BAMBI*) were commonly overexpressed in all three studies, as were the genes determined as noteworthy within the KEGG analysis, such as *GJA1* and *LEF1*.

### 3.2. Immunohistochemical Characterisation

#### 3.2.1. H-Score Analysis

*ASPN*, *STK3* and *BAMBI* were all significantly overexpressed at the mRNA level in both canines (*p* = 0.02175, *p* = 0.00985, and *p* = 0.032, respectively) and humans (*p* = 0.000978, *p* = 0.019706, and *p* = 0.010047, respectively) OSA tumours were compared to non-malignant counterparts identified in previous research [43,44]. In addition, qRT-PCR validated the RNASeq results. *ASPN* showed elevated expression in canine OSA tumours compared to non-malignant counterparts (n = four matched samples; *p* = 0.0091) [43]. *STK3* and *BAMBI* qRT-PCR results validated significant overexpression of the genes in tumour compared to patient-matched, non-tumour tissues (n = seven matched pairs of samples, *p* ≤ 0.05 for both genes). Hence, these genes were selected for protein expression investigation via immunohistochemical techniques.

A summary of the IHC staining results for the three respective proteins of interest is displayed in Table 1 and Table 2. Overall, all antibodies displayed diffuse staining across OSA tissue samples (n = 19), with most exhibiting positive stromal staining to varying degrees of intensity (Figure 2A, Figure 3A and Figure 4A). In further detail, H-score analysis for ASPN, STK3 and BAMBI was performed for score distribution and correlations between nuclear and cytoplasmic H-scores, as well as for investigating anatomical location and sex differences in subcellular staining (Figure 2, Figure 3 and Figure 4).

#### 3.2.2. ASPN H-Score Analysis

OSA tissues (n = 18) displayed minimal positive ASPN staining across most tissue samples. Negative nuclear staining was exhibited in 3/18 samples (16.67%). Cytoplasmic and nuclear scores were compared, showing no statistical difference between the subcellular locations (*t*-test, *p* = 0.7817; Figure 2B). A total of 2/18 (11.11%) specimens had both negative nuclear and cytoplasmic staining, whilst one (1/18; 5.56%) specimen displayed negative nuclear and low cytoplasmic staining (Figure 2C). The remaining 15/18 samples exhibited both nuclear and cytoplasmic ASPN staining to some degree of intensity. The majority of samples showed low nuclear and low cytoplasmic staining (14/18; 77.78%), and 1/18 (5.56%) samples displayed higher nuclear and cytoplasmic staining compared to the other specimens presented in this study (Figure 2D). Overall, staining was diffuse, and stromal staining was visible in 86.67% of the photomicrographs scored. There proved to be a strong positive correlation between nuclear and cytoplasmic mean H-scores (R^2^ = 0.9836; Figure 2D). Anatomical location and sex differences for both nuclear and cytoplasmic mean H-scores was also investigated, but *t*-test results showed that there were no significant differences between these variables (Figure A1B).

#### 3.2.3. STK3 H-Score Analysis

STK3 staining was diffuse, and there was a positive STK3 stromal presence in 84.21% of n = 19 samples. There was a statistically significant difference between nuclear and cytoplasmic H-scores (*t*-test, *p* ≤ 0.0001; Figure 3B). STK3 positive nuclear staining was observed in 13/19 (68.42%) specimens at low levels, whilst the remaining 6/19 (31.58%) tissues exhibited no nuclear staining (Figure 3C). Cytoplasmic STK3 staining was displayed in 18/19 (94.74%) samples with varying intensity. STK3 cytoplasmic H-scores were low in 7/19 (36.84%), moderate in 8/19 (42.11%) and high in 3/19 (15.79%) of specimens (Figure 3C). 1/19 (5.26%) specimens displayed low nuclear staining and absent cytoplasmic staining (Figure 3C). There was a weak positive correlation between nuclear and cytoplasmic STK3 expression (R^2^ = 0.1918; Figure 3D). Differences in nuclear and cytoplasmic mean H-scores were investigated comparing axial and appendicular locations, as well as sex differences; *t*-test results showed there were no significant differences between these variables (Figure A1C).

#### 3.2.4. BAMBI H-Score Analysis

Overall, BAMBI staining was diffuse, and stromal staining was evident in 82.22% of n = 18 tissue samples used in this study. There was a significant difference in nuclear and cytoplasmic H-scores (*t*-test, *p* ≤ 0.0001; Figure 4B). Positive BAMBI expression was observed in the cytoplasm of all samples at varying degrees of intensity. Moderate cytoplasmic staining was exhibited in the majority of samples (10/18; 55.56%), with some displaying high expression (5/18; 27.78%), and the remaining with low (3/18; 16.67%) (Figure 4C). Nuclear staining, however, remained consistently low or absent (13/18; 72.22% and 5/18; 27.78%, respectively) in all specimens used in this study (Figure 4C). A weak positive correlation was observed between nuclear and cytoplasmic H-scores (R^2^ = 0.1878; Figure 4D). Nuclear and cytoplasmic mean H-scores were also compared to identify sex and anatomical location differences between the variables, which concluded that there were no significant differences (Figure A1D).

## 4. Discussion

Connexin 43 (or GJA1) facilitates intercellular and extracellular environment communication via the formation of gap junctions and hemichannels [52], allowing the passage of molecules less than 1.2 kDa into cells [53]. In the present study, Connexin 43 was a commonly overexpressed gene (canine; *p* = 0.0002 and human; *p* ≤ 0.0001) [43,44]. There is evidence supporting its dysregulation within OSA tumours and its role as a tumour suppressor; however, there is disagreement on whether it is over- or underexpressed within primary neoplasms. Notably, previously published work investigated cell lines, whereas the present research investigated patient tumours. Previous studies seeking to upregulate GJA1 using the Coleusin factor (human OSA cells) [53,54,55] or astaxanthin (human and animal cells) [56] may not, therefore, be applicable in vivo. Moreover, although astaxanthin reduced cell proliferation in canine cells in vitro, it did not affect GJA1 expression in canine OSA [57]. Our results, compared to in vitro studies, indicate that studies investigating GJA1 reduction in vivo are required.

Cyclin D2 (*CCND2*) and cyclin-dependent kinase inhibitor 1A (*CDKN1A,* gene for P21) have been identified within the p53 signalling pathway. In the present study, *CCND2* was significantly overexpressed (*p* = 0.01855) in canine but not human OSA, whereas *CDKN1A* was only significantly underexpressed in human OSAs (*p* ≤ 0.0001). The canine overexpression of *CCND2* concurs with other studies as this oncogene is overexpressed in numerous cancer types [58]. By contrast, our human OSAs showed no significant *CCND2* changes. The microRNA, miR-1297, targets CCND2 directly, resulting in proliferation inhibition, G1 phase arrest and reduced tumour growth in OSA [58], and miR-2682-3p also targets CCND2 in OSA [59]. These microRNAs may facilitate the early detection of OSA [60], and CCND2 and/or microRNAs hold potential as therapeutic targets in canine patients but may not be as applicable in humans. P21 loss of expression has previously been observed in OSA cell lines and tissues [61,62], similar to *CDKN1A* results observed in our human, but not canine, OSAs. Increasing its levels in human OSA would potentially decrease neoplastic cell proliferation.

The present study identified *LEF1* (lymphoid enhancer-binding factor 1) as a common DEG that is significantly elevated in canine and human OSA (*p* = 0.0405 and *p* =< 0.0001, respectively). *LEF1* was also involved in 7 of the 10 most commonly overexpressed KEGG pathways. Although none of these pathways were deemed significant, this highlights the importance of LEF1 and its potential role as a biomarker and drug target in OSA. We previously discussed the current landscape of drugs targeting the Wnt/β-catenin/LEF1 pathway in cancer, including the challenges involved in successfully targeting this pathway due to its crucial physiological role in the body [51]. Since the publication of our previous paper, studies targeting β-catenin using a small molecule inhibitor ICG001, or knocking down LEF1, have resulted in the reversal of Lenvatinib resistance in hepatocellular carcinoma cells [63]. To the best of our knowledge, no new drugs targeting the Wnt/β-catenin/LEF1 pathway have been discovered, nor have any existing inhibitors in trials been clinically approved for treating cancer.

This present study identified *NQO1* RNA as being significantly underexpressed in canine and human OSA (*p* = 0.00975 and *p* ≤ 0.0001, respectively). NAD(P)H:quinone oxidoreductase is an enzyme encoded by the *NQO1* gene. NQO1 plays a role in cell death [64], and absent or lowered NQO1 has been associated with increased susceptibility to cancer; however, NQO1 is upregulated in many cancers, including breast, pancreatic and colon cancer [64]. The chemotheraputic agent cisplatin is commonly used to treat OSA in both species and significantly upregulates NQO1 in cancer cells, increasing their sensitivity to β-lapachone in vitro and in vivo [65]. Napabucasin (NAPA) has also been used on human OSA cell lines, decreasing cell viability and inducing apoptosis, and inhibiting metastatic progression in vivo [66]. However, there is evidence to suggest that NQO1 is a major determinant of NAPA efficacy [67]. NAPA acts as a substrate for NQO1, facilitating the generation of reactive oxygen species that lead to cell death [67]. NQO1 has also been linked to OSA via p53 mutations, leading to significantly lower survival rates [68,69] and early-life OSA and Li-Fraumeni syndrome, a predisposing factor of OSA and other sarcomas in people [70,71]. Normal p53 function is pivotal in preventing neoplastic MSC transformation [72,73,74], and NQO1 has protective roles for binding to p53 and increasing protein stability whilst inhibiting proteasomal degradation in response to DNA-damaging stimuli. This research that indicates NQO1 expression could act as a companion diagnostic test if considering NAPA-based regimens for cancer therapeutics, including OSA.

We previously identified that *ASPN* (asporin/*PLAP-1*) mRNA was higher in canine OSA tumours as compared to normal bone (*p* = 0.02175) [43], and the present study exhibited human OSA overexpression (*p* = 0.000978). Little is known about the role of ASPN in canine or human OSA [44]. ASPN belongs to a family of small leucine-rich proteoglycan (SLRPs) proteins [75,76]. It negatively regulates chondrogenesis in vitro by inhibiting TGF-β function [77], and plays a role in bone and joint diseases, such as osteoarthritis [78,79]. ASPN inhibition may restore normal collagen fibrillogenesis and extracellular matrix organisation and attenuate pro-oncogenic signalling, including TGF-β. A phase I/II OSA trial using the TGF-βR1 inhibitor vactosertib is ongoing (NCT05588648). Our results also indicate that targeting ASPN may also present an alternative approach to targeting TGF-β signalling for OSA treatment in both people and dogs.

The results of the present study remained consistent with the literature regarding the presence of positive stromal staining [80]. Positive nuclear and cytoplasmic staining were also present in most canine samples, similar to testis, adipose tissues, breast and soft tissues in the Human Protein Atlas. ASPN regulates several signalling pathways in the tumour microenvironment (TME) [81,82]. It is thought that the secreted form of ASPN inhibits TGF-β signalling, whereas cytoplasmic ASPN promotes TGF-β signalling through interaction with Smad2/3 [79,83]. In OSA, TGF-β signalling is largely considered to have pro-tumorigenic effects, including the promotion of metastasis; thus, the cytoplasmic role of ASPN in TGF-β signalling may be the underlying mechanism [84]. As staining in the present study was generally weak in canine OSA, optimizations are required to advance H-scoring in clinical settings. Due to the high percentage of canine specimens displaying stromal ASPN (86.67%), combined with the lower scores within subcellular locations in the present study, it may be possible that elevated ASPN secretion levels are via cancer-associated fibroblasts (CAFs) within the TME. ASPN can be expressed by CAFs, promoting cancer cell invasion and metastasis, whilst the knockdown of ASPN in CAFs reduced fibroblast and gastric cell invasion in vivo [85,86]. CAFs play important roles in the progression, metastatic spread and drug resistance of OSA [87], though their secretion of ASPN and its potential impact on tumourigenesis have yet to be investigated in this cancer type, representing an area of future research.

The STRING analysis of ASPN in *Canis lupus familiaris* (accessed 10.03.2025) in the present study revealed its co-expression with periostin (POSTN), a secreted extracellular matrix protein involved in cancer stem cell maintenance and the metastatic progression of cancers [88]. In human OSA, POSTN was highly expressed and associated with poor disease prognosis [89]. *POSTN* was also identified as significantly overexpressed at the mRNA level (*p* = 0.03365) in canine OSA, suggesting it may also be involved in the metastatic progression of canine OSA [43]. The STRING database also revealed co-expression of ASPN with other SLRPs, osteoglycin (OGN) and osteomodulin (OMD), which have similar protein homologies and play pivotal roles in cellular growth, differentiation and metastasis. *OGN*, *OMD* and *ASPN* can be arranged in tandem clusters on the human chromosome 9q2 [90]. Studies need to be conducted in canids to determine whether *ASPN*, *OGN* and *OMD* are clustered on the same chromosome in dogs, and whether this influences their co-expression. The present study, and the wider literature, provides evidence that ASPN supports the stroma, and future studies should be conducted to determine whether these mechanisms promote the invasion, oncogenesis and progression of OSA.

Currently, little is known about STK3’s role in canine or human OSA. KEGG analysis in this OSA study identified the Hippo signalling pathway as significant in canine, but not human, OSA. In addition, the present study showed that protein expression was present in the stroma and nucleus, and significantly higher in the cytoplasm. Serine/threonine kinase 3 (STK3) is a protein encoded by the *MST2*/*STK3* gene, which was overexpressed in canine and human OSA within this study (*p* = 0.00985 and *p* = 0.019706, respectively) [43,44]. STK3 is a key enzymatic component within the Hippo signalling pathway (Figure A2A) that is vital for controlling tumour suppression via the restriction of cellular proliferation and the promotion of apoptosis [91]. The Hippo/YAP pathway has been intensively researched as it operates at various stages of tumour growth, including primary tumour development, angiogenesis, the epithelial-to-mesenchymal transition (EMT) and dissemination to secondary sites [92]. STK3 also has roles in the regulation of immune cell infiltration [93] and modulation [94]. For example, STK3 upregulation correlated with the chemotaxis of CD8+ T-cells via increasing production of chemokines CXCL16 and CX3CL1 [94]. RNA sequencing analysis in this study revealed a significant overexpression of *CX3CL1* (*p* = 0.0276) in canine OSA, and the high percentage of positive STK3 stromal staining also supports its potential role in mediating TME immune signalling. However, it is thought that CD8+ T-cell expression in OSA is suppressed by invading tumour cells [95]; therefore, further work needs to investigate whether elevated STK3 expression has the capability to mediate immune cell infiltration in this cancer type. The advances in the present study indicate that STK3 may be a novel biomarker and a therapeutic target in OSA.

STK3 expression dysregulation has also shown to have pro-oncogenic effects in different cancer types. In prostate cancer, STK3 was regularly elevated and associated with decreased overall survival of patients [96] and increased expression in breast cancer, which is correlated with worse patient outcomes [97]. The STK3/4 small-molecule inhibitor XMU-MP-1 displayed growth inhibition and reduced proliferation in prostate cancer cell lines [96]. It also resulted in reduced MOB1 and YAP phosphorylation in both prostate and breast cancer cell lines and 3D models [96]. STK3 variants, present in 3.2–15% of several cancer types, resulted in impaired kinase and suppressive capabilities within the Hippo signalling pathway [98]. The present study did not investigate the new STK3 variant in OSA, but this may be an interesting avenue to explore. Targeting STK3 may, therefore, provide a new therapeutic approach for both canine and human OSA due to its overexpression in both species.

Canine and human *STK3* levels were significantly overexpressed in the present study, which is important given that obesity is a risk factor for many cancers, including OSA. STK3 has shown to be a key suppressor of mitochondrial capacity in adipose tissues with high concentrations of STK3, which were observed in white adipose fat as opposed to thermogenic, brown fat [99]. Elevated levels of STK3 have been shown to correlate with obesity in humans, with pharmacological inhibition in mouse models showing improvements in metabolic profiles of these animals [99]. Molecular dysregulation of the Hippo signalling pathway and its influence on the pathogenesis of metabolic diseases, such as obesity, may also provide an exciting avenue for the developments of targets in adipocyte homeostasis for the prevention of OSA [100]. Further work on adipogenesis, including adipocyte composition and quantity, may provide more information on the metabolic status of OSA, informing drug development/repurposing and its consequent efficacy.

In the present study, *BAMBI* was significantly overexpressed in both canine and human OSA (*p* = 0.032 and *p* = 0.010047, respectively). The BMP and activin membrane-bound inhibitor (*BAMBI*) is an evolutionary conserved gene that encodes a transmembrane protein related to the type I receptors of the TGF-β family [43,44]. BAMBI expression is increased in many carcinomas, contributing to the proliferation and metastasis of neoplastic cells due to escape from TGF-β growth arrest (Figure A2B) [101]. IHC in human OSAs [102] demonstrated cytoplasmic BAMBI expression, similarly to the present canine study, which showed significantly higher expression within the cytoplasm compared to the nucleus (*p* < 0.0001); cytoplasmic staining was also observed in more samples compared to nuclear. Overexpression of BAMBI in human OSA cell lines resulted in a more invasive and migratory phenotype [102], and correlated with an upregulation of matrix metalloproteinases (MMPs), including MMP-2 and MMP-9, which are involved in metastatic dissemination and invasion [102]. The study also showed roles in promotion of cell growth and as a potential regulator of Wnt and catenin-induced target genes involved in cell cycle promotion, such as CCND1 (cyclin D1) and protein-dependent kinases [102]. It is worth noting that, in the present study, *CCND1* was not differentially expressed in canine tumours and was significantly underexpressed in human OSA (*p* = 0.008187). miR-93-5p, an effector of the TGF-β signalling pathway in prostate cancer, has been predicted to act on and bind to *BAMBI*, consequently inhibiting *BAMBI* expression and activating TGF-β signalling [103]. Our results support that BAMBI represents a target for gene therapy, and the use of microRNAs may prove to be a novel therapeutic approach in canine and human OSA.

Whilst many large breeds of dogs are predisposed to OSA, this study focused on Rottweilers, in line with the mRNA sequencing results. Biomarker expression differences may vary between breeds; therefore, findings could be validated in other breeds that are highly predisposed to OSA. Undertaking staining in non-tumour bone may also highlight differential staining locations between control and tumour tissues. A larger number of appendicular tumours were used in this study, since this represents the more common anatomical location of OSA; however, further studies into appendicular versus axial locations, alongside age-matched male and female samples, should be performed in the future. The sample size of n = 19 enabled an in-depth understanding of protein expression, ensuring that statistical analysis could be performed. Future studies may benefit from undertaking full clinical and histopathological studies including grade of tumour, BMI/body condition score, concurrent medical conditions, neutering status and comparing H-scores in human OSA.

The present study used a meta-analysis comprised of eight different GEO datasets, containing combinations of primary and secondary tissues and cell lines, to compare against the canine RNASeq data. We acknowledge that there are many other available datasets; hence, additional analysis was performed comparing another human dataset to the initial DEG analysis. In human OSA, work has started on other potentially important features, such as satellite and repetitive elements, which represent interesting future avenues for OSA [104,105]. Future gene expression, KEGG pathway analysis and histological studies may also compare the differences in gene expression pre- and post-treatment. IHC and H-scoring are standardised diagnostic tools used within human medicine but have not yet been implemented in most veterinary diagnostic settings. Research, like this present study, is required prior to the implementation of IHC and H-scoring into veterinary clinics, as biomarkers need to be identified and validated. Advances in AI may also increase the affordability and accessibility of these diagnostic and prognostic techniques and expand the research into treatment opportunities for not only canine and human OSA, but for other cancer types in all species.

## 5. Conclusions

This present study identified common DEGs in human and canine OSA tumours compared to non-malignant counterparts, highlighting the similarities and differences between these species. Targeting these DEGs/pathways may provide insights into potential treatment options for the disease. The shorter lifespan of dogs, higher incidence of OSA and natural rapid progression of the disease allows canine models to be used to progress and develop novel therapeutic protocols that can be translated for use in people. The results from this investigation emphasise the potential use of various markers that could facilitate diagnosis, prognosis and treatment opportunities for OSA. Presently, IHC techniques are used extensively and routinely in human medicine to aid in the diagnosis and prognosis of OSA; however, this research accentuates their relevance in veterinary medicine. Deeper investigations into commonly overexpressed genes, including *ASPN*, *STK3* and *BAMBI*, via immunohistochemistry revealed novel findings on cellular protein expression in canine tumours. Given the staining observed, their significant subcellular staining locations and their involvement in signalling pathways, including TGF-β and Hippo identified via KEGG analysis, STK3 and BAMBI are promising biomarkers and targets for future OSA therapies. Although ASPN staining results were not deemed significantly different in terms of subcellular location, its involvement in the TGF-β signalling pathway may provide interesting avenues for future therapeutic regimens. A deeper understanding of genetic and protein interactions, and consequently their mechanisms within cancer development, is required to further enhance the progression of novel diagnostic, prognostic and treatment opportunities in canine and human OSA.

## Figures and Tables

**Figure 1 life-15-00951-f001:**
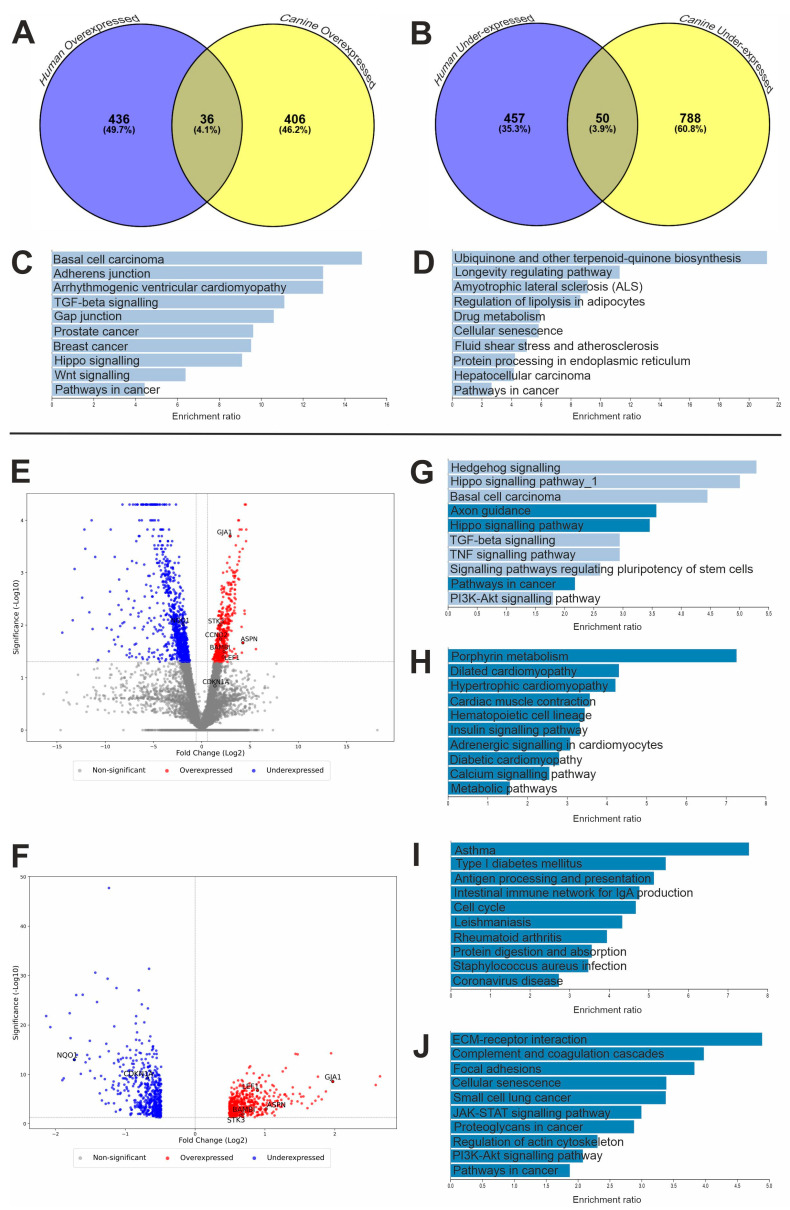
Genetic analysis results. Venn diagrams displaying common DEGs in canine and human OSAs. Number of common (**A**) over- and (**B**) underexpressed genes between canines and people. Over-representation KEGG pathway analysis of shared DEGs between human and canine OSA; common pathway analysis results for (**C**) overexpressed genes (n = 36) and (**D**) underexpressed genes (n = 50). The pathways shown did not reach significance (FDR-corrected *p*-value < 0.05). Over-representation KEGG pathway analysis of DEGs (**E**) canine and (**F**) human OSA DEG volcano plots; genes highlighted within the discussion are annotated. Canine KEGG analysis: (**G**) over- and (**H**) underexpressed. Human KEGG analysis: (**I**) over- and (**J**) underexpressed. (**C**,**D**,**G**–**J**) Webgestalt KEGG pathway analysis. Significant pathways depicted in dark blue (FDR < 0.05), and non-significant pathways depicted in light blue (FDR ≥ 0.05).

**Figure 2 life-15-00951-f002:**
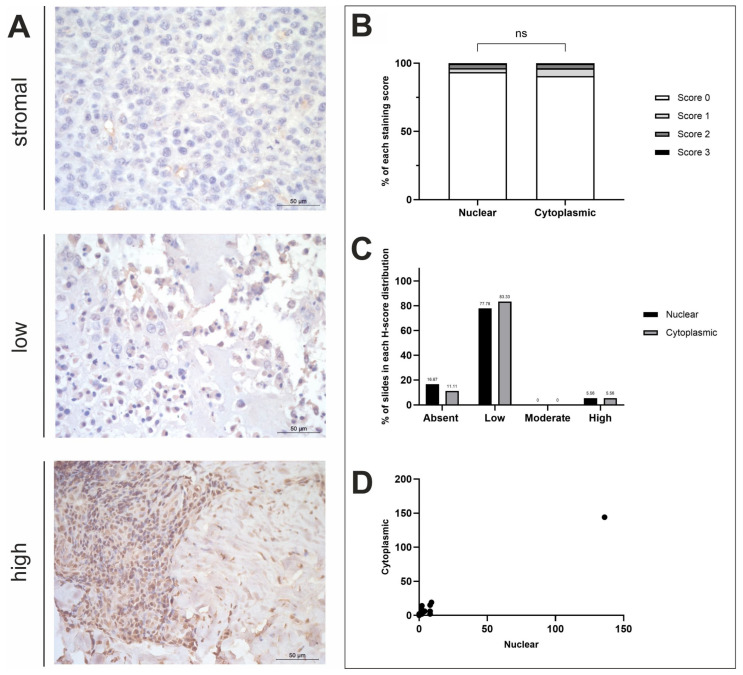
ASPN H-score analysis. (**A**) ASPN immunohistochemistry photomicrographs obtained at ×400 fold magnification depicting positive protein expression at varying intensities (top row = stromal, middle row = low, bottom row = high). Scale bars represent 50 µm. (**B**) Average nuclear and cytoplasmic H-scores (scores 0–3; n = 18). (**C**) Distribution of nuclear and cytoplasmic H-scores (groups calculated in thirds of the largest range of scores); low (≤48), moderate (49–96), high (≥97). (**D**) Nuclear and cytoplasmic H-scores. ns = *p* ≥ 0.05.

**Figure 3 life-15-00951-f003:**
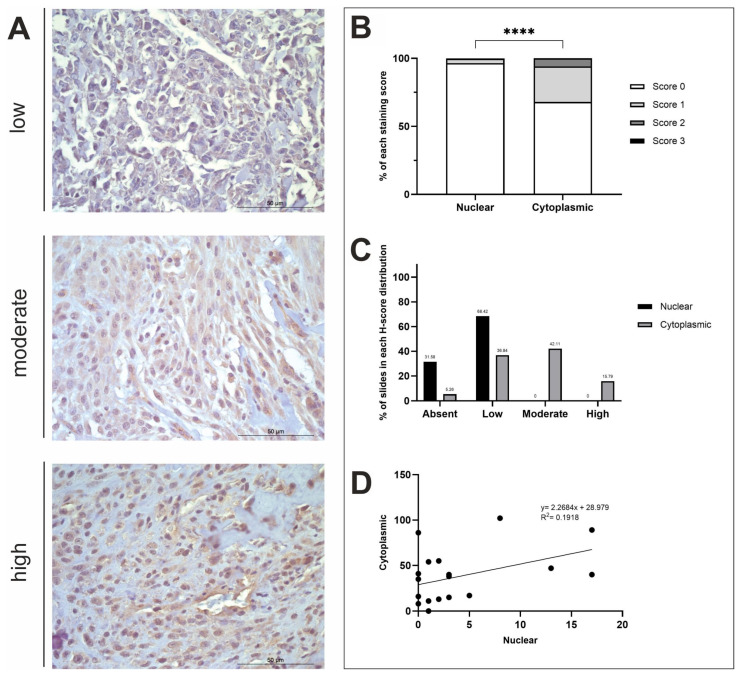
STK3 H-score analysis. (**A**) STK3 immunohistochemistry photomicrographs obtained at ×400 fold magnification depicting positive protein expression at varying intensities (top row = low, middle row = moderate, bottom row = high). Scale bars represent 50 µm. (**B**) Average nuclear and cytoplasmic H-scores (scored 0–3; n = 19). (**C**) The distribution of nuclear and cytoplasmic H-scores (groups calculated using a third of the largest range of scores); low (≤34), moderate (35–68), high (≥69). (**D**) The correlation between nuclear and cytoplasmic H-scores. **** *p* < 0.0001.

**Figure 4 life-15-00951-f004:**
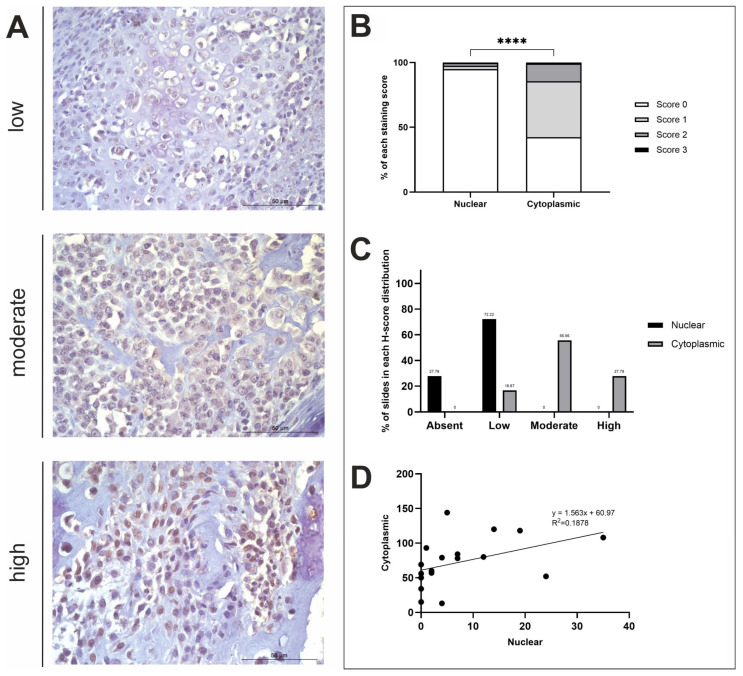
BAMBI H-score analysis. (**A**) BAMBI immunohistochemistry photomicrographs obtained at ×400 fold magnification depicting positive protein expression at varying intensities (top row = low, middle row = moderate, bottom row = high). Scale bars represent 50 µm. (**B**) Average nuclear and cytoplasmic H-scores (scored 0–3; n = 18). (**C**) The distribution of nuclear and cytoplasmic H-scores (groups calculated using a third of the largest range of scores): low (≤43), moderate (44–86), high (≥87). (**D**) The correlation between nuclear and cytoplasmic H-scores. **** *p* < 0.0001.

**Table 1 life-15-00951-t001:** Subcellular staining H-scores (nuclear and cytoplasmic).

Nuclear	**Cytoplasmic**				
		**Absent**	**Low**	**Moderate**	**High**
	[ASPN, n = 18]				
	Absent	2 (11.11%)	1 (5.56%)	-	-
	Low	-	14 (77.78%)	-	-
	Moderate	-	-	-	-
	High	-	-	-	1 (5.56%)
	[STK3, n = 19]				
	Absent	-	3 (15.79%)	2 (10.53%)	1 (5.26%)
	Low	1 (5.26%)	4 (21.05%)	6 (31.58%)	2 (10.53%)
	Moderate	-	-	-	-
	High	-	-	-	-
	[BAMBI, n = 18]				
	Absent	-	2 (11.11%)	3 (16.67%)	-
	Low	-	1 (5.56%)	7 (38.89%)	5 (27.78%)
	Moderate	-	-	-	-
	High	-	-	-	-

Percentage of specimens showing mean nuclear and cytoplasmic H-scores within each of the four distribution categories (absent, low, moderate or high) for each protein encoded by the three genes of interest (*ASPN*, *STK3* and *BAMBI*).

**Table 2 life-15-00951-t002:** H-scores for ASPN, STK3 and BAMBI.

Protein (No. of Cases)	Staining Distribution (Diffuse/Multifocal/Focal)	Stromal Staining (% of Slides)	Cellular Location	H-Score	
				Mean ± SEM (2 s.f)	Range (Min–Max)
ASPN (n = 18)	Diffuse	86.67%	Nuclear Cytoplasmic	10.56 ± 3.30 13.56 ± 3.50	0–165 0–155
STK3 (n = 19)	Diffuse	84.21%	Nuclear Cytoplasmic	4 ± 1.29 38.05 ± 6.69	0–17 0–102
BAMBI (n = 18)	Diffuse	82.22%	Nuclear Cytoplasmic	7.56 ± 2.31 72.78 ± 8.35	0–35 13–144

H-scores for both the nuclear and cytoplasmic staining for ASPN, STK3 and BAMBI. Expressed as mean and standard error of the mean (SEM) values, H-score ranges (nuclear and cytoplasmic), in addition to staining distribution and percentage of slides with stromal staining.

## Data Availability

Data is available upon request from the corresponding authors. The data are not publicly available due to being part of an ongoing study.

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
