# Peer review of "Comparison of Differentially Expressed Genes in Human and Canine Osteosarcoma"

_life, 2025, doi:10.3390/life15060951_

Round 1
Reviewer 1 Report
Comments and Suggestions for Authors
The manuscript entitled "Comparison of differentially expressed genes in human and canine osteosarcoma" by Jackson-Oxley et al. describes an original scientific report aimed at investigating the differential expression profile of canine and human osteosarcoma.
The paper is primarily based on published RNAseq data from which the authors extracted data. From this list of differentially expressed genes, they analysed a number of proteins by immunostaining in canine osteosarcoma.
The present work is really difficult to follow and the main aim of the work is not explained (apart from the fact to generate this DEG list of genes). For example, the IHC performed is not comparative and therefore does not really provide any further information other than the fact that it is more or less expressed in OS.
Main point : The authors discuss the interest of having access to the canine OS to understand the human pathology, but from their data only 36 genes out of 436 and 406 were commonly upregulated and 50 out of 457 and 788 downregulated. This very limited number of common genes raises significant questions about the 'interest' of canine OS in addressing human pathology. Why did the authors not functionally investigate these genes in human OS?
- The relationship between the IHC data and the bio-information is not rationally explained.
Author Response
Reviewer 1
Comments 1: The manuscript entitled "Comparison of differentially expressed genes in human and canine osteosarcoma" by Jackson-Oxley et al. describes an original scientific report aimed at investigating the differential expression profile of canine and human osteosarcoma.
The paper is primarily based on published RNAseq data from which the authors extracted data. From this list of differentially expressed genes, they analysed a number of proteins by immunostaining in canine osteosarcoma.
Response 1: Thank you for your summary and thank you for your time and effort reviewing our research.
Comments 2: The present work is really difficult to follow and the main aim of the work is not explained (apart from the fact to generate this DEG list of genes). For example, the IHC performed is not comparative and therefore does not really provide any further information other than the fact that it is more or less expressed in OS.
Response 2: Thank you, the overarching aim was to look at human versus canine OSA data but we have now explained the IHC and H-scoring a little better (lines 154-174). We have added a section (see lines 148-157) in the introduction highlighting the aims of the research. From the IHC and the H-scoring results we were able to confirm that the genes which were overexpressed in the mRNA sequencing data set were also being translated into proteins, as this is not always the case. In addition, we looked at their cellular (nuclear and cytoplasmic) expression and used quantitative H-scoring to give insights into protein expression, localisation and provide further avenues using IHC and H-scoring in diagnosis, prognosis and drug development.
Comments 3: Main point: The authors discuss the interest of having access to the canine OS to understand the human pathology, but from their data only 36 genes out of 436 and 406 were commonly upregulated and 50 out of 457 and 788 downregulated. This very limited number of common genes raises significant questions about the 'interest' of canine OS in addressing human pathology. Why did the authors not functionally investigate these genes in human OS?
Response 3: Whilst there were few genes that were common DEGs between the two species, it is known that canine OSA is used as a model for the disease in humans, especially for paediatric cases due to shared similarities such as metastatic progression, the presentation of the disease, bimodal age distribution of affected patients in both species, origins of disease, treatment opportunities etc. This study looked deeper into the genetic makeup of the tumours and identified pathways that may be of interest when investigating future drug opportunities that may be applicable to both species. Druggable targets have been suggested from this as well as information on potential therapeutic agents discovered within the literature that may pose beneficial for human OSA and canine OSA alike. Canine tissues were then used to show subcellular protein expression of three selected commonly expressed DEGs to assess whether the protein expression occurred in the tissues. This also showed the applicability of using H-scoring and immunohistochemical techniques in the veterinary field as well as in human medicine. It is noted that human tissues could have been used for this also, however this study is also expanding on previous research conducted by the lab group who have an extensive background in canine OSA tissue use and have access to these tissues.
Comments 4: The relationship between the IHC data and the bio-information is not rationally explained.
Response 4: Thank you, we have now provided greater detail on this. Please see lines 158-174, 804-839, and our previous responses too.
Reviewer 2 Report
Comments and Suggestions for Authors
This is a great manuscript, and it adds lots of new knowledge on osteosarcoma. The comparison between canine and human OS is a great approach.
However, I have many concerns before the manuscript can be published:
1. What is the sample size? Please state it clearly in the abstract and the methods.
2. Many overlapping genes between humans and dogs on OS were found. Were the results adjusted to the effect of the treatment? What about the treatment effect? Are the drugs the same for dogs and humans?
3. The authors do not mention or discuss other previously published studies, like PMID: 25496518 and PMID: 29050494, which presented a whole genome transcriptome analysis and even a combination of DNA and RNA analysis. How are their results compared to those of these studies?
4. The authors need to give more background information. Again, what about the repetitive and satellite elements that are changed in many cancers and OS (PMID: 29250102)?
5. The authors used already published data but did not use one of the most significant OS transcriptomic studies (PMID: 38966281), which was recently published and has data available. The study PMID: 38966281 is a large experimental work that combined different experiments and built a complex OS gene expression footprint.
Author Response
Comments 1: This is a great manuscript, and it adds lots of new knowledge on osteosarcoma. The comparison between canine and human OS is a great approach.
Response 1: Thank you very much, we are delighted you like our work and thank you for your time and effort reviewing our manuscript.
Comments 2: However, I have many concerns before the manuscript can be published:
1. What is the sample size? Please state it clearly in the abstract and the methods.
Response 2: The sample size for IHC conducted in this study alongside information regarding the sex and location is stated in the methods section 2.2.1 (lines 238-247). The n number for each of the proteins investigated is also noted in Tables 1 and 2 (lines 387 and 392) as well as in the results sections 3.2.2, 3.2.3, and 3.2.4 (lines 402, 406, 457, 461, 480 and 484). We have now added a section into the abstract (line 34) further highlighting the sample size for this study.
Comments 3: Many overlapping genes between humans and dogs on OS were found. Were the results adjusted to the effect of the treatment? What about the treatment effect? Are the drugs the same for dogs and humans?
Response 3: The canine samples used for the mRNA sequencing analysis were treatment naïve as they were excised from patients following amputation surgeries. The human samples from the mRNA sequencing conducted by Yang et al. 2014 were from a combination of samples excised from bone tissues and cell lines (both primary and metastatic disease). Most of these samples were treatment naïve, however there were samples where it was not stated as to whether they were collected pre or post treatment. This is something that we have considered as it will impact the genes expressed but interestingly few studies state treatment stage or consider it. Future studies may involve investigating differences in gene expression pre and post treatment and investigating the pathways affected by this, as well as addressing whether these changes are having positive effects at reducing OSA development. Currently, the treatment protocols for both species are similar (see section 1.2, lines 88-143), though new chemotherapeutic agents are deemed necessary for the treatment of OSA in both species, hence the comparative DEG analysis and pathway analysis in this study, as well as previous work conducted by our lab group.
Comments 4: The authors do not mention or discuss other previously published studies, like PMID: 25496518 and PMID: 29050494, which presented a whole genome transcriptome analysis and even a combination of DNA and RNA analysis. How are their results compared to those of these studies?
Response 4: We have added in PMID: 29050494 to the methods and added PMID: 25496518 into the discussion in the limitations section.
Comments 5: The authors need to give more background information. Again, what about the repetitive and satellite elements that are changed in many cancers and OS (PMID: 29250102)?
Response 5: We have now noted this in the discussion as future research (lines 824-826) and cited that paper as an example. Given the depth and breadth of information this may be something for future research in canine OSA.
Comments 6: The authors used already published data but did not use one of the most significant OS transcriptomic studies (PMID: 38966281), which was recently published and has data available. The study PMID: 38966281 is a large experimental work that combined different experiments and built a complex OS gene expression footprint.
Response 6: Thankyou for pointing this out. We originally chose the analysis we used in this study as it is a META analysis combining 8 different GEO datasets (it has OSA samples from cell lines, bone tumours, as well as metastatic disease). This gives a comprehensive oversight compared to just using one individual dataset. We have done additional analyses and identified the common over- and under- expressed DEGs between the initial analysis (Yang et al. v Simpson et al.) and the new analysis using 89/90 samples used in the PMID: 38966281 study and we have cited this study. Data was available for all but one of their samples. Our further analysis showed that 33/36 overexpressed DEGs were common significant genes between all three studies, and 38/50 were also common under expressed DEGs between the three studies. Additionally, the three genes which underwent immunohistochemical investigations for their protein expression in canine OSA tissues (ASPN, STK3, and BAMBI) were all commonly overexpressed between all three studies. This has now been added into the paper.
Reviewer 3 Report
Comments and Suggestions for Authors
The study focusses on DEG and their products in osteosarcoma of dogs and humans. Canine samples were investigated in detail immunohistologically. The immunohistology results might be very helpful for validation of the dog as a model to understand osteosarcoma.
The introduction is long but very informative.
The abstract does not concisely reflect the results of the study (lines 34-35).
Method: it is very good that only Rottweilers are included. H scoring method. Was a semiquantification (densitometry) not possible due to heterogenicity of the tissue?
Figure 1: the text in the figure is not readable
Figures 2-4: please add the information that canine tissue was investigated in the legends. mention the source/donor of the tissue. are really 40x fold or rather 400x fold magnifications shown?
line 352: "mod" should be explained as moderate
Discussion: The discussion section explains the targets investigated very well. The limitations of the study could be presented. Could tissues (bone?) be included into the analysis with basal expression of the respective DEG products? Were the different locations (axial/appendicular) compared? There is distinguished between nuclear and cytoplasmic expression of the targets but the interpretation is not clear.
Minor: blanks are lacking or surplus (e.g., lines 311 versus 313 etc., 351), line 455 versus 463 connexin should either start with capital or normal sized letter. The abbreviation OSA is not consistently used.
Author Response
Comments 1: The study focusses on DEG and their products in osteosarcoma of dogs and humans. Canine samples were investigated in detail immunohistologically. The immunohistology results might be very helpful for validation of the dog as a model to understand osteosarcoma.
Response 1: Thank you very much for your positive review and comments and thank you for your time and effort reviewing our manuscript.
Comments 2: The introduction is long but very informative.
Response 2: Thank you. The other reviewers also liked the detailed introduction so we have retained is. We added a few more words to the aims only to clarify for reviewer 1. We recognise it is relatively long but we needed to explain both the canine and human OSA situation and a number of techniques including some that are not yet widely used in OSA histopathology or veterinary medicine.
Comments 3: The abstract does not concisely reflect the results of the study (lines 34-35).
Response 3: We have added additional information into the abstract (lines 32-37) to address the IHC analysis in this study, including investigating sex and anatomical location differences of nuclear and cytoplasmic protein expression for the three proteins investigated within the study. Due to restraints on the word limit, we have just summarised what has been completed as sadly there is not enough room to detail every result.
Comments 4: Method: it is very good that only Rottweilers are included. H scoring method. Was a semi quantification (densitometry) not possible due to heterogenicity of the tissue?
Response 4: Thank you very much, yes the tissue is extremely heterogeneous therefore H-scoring is better and also could play a large role in future clinical histopathology.
Comments 5: Figure 1: the text in the figure is not readable
Response 5: Thank you for highlighting this, the requested format in word does not give an especially high resolution but now we have been able to upload all figures in a high resolution which means the reader will be able to see them and we hope you will have access to the higher resolution images too as a reviewer.
Comments 6: Figures 2-4: please add the information that canine tissue was investigated in the legends. mention the source/donor of the tissue. are really 40x fold or rather 400x fold magnifications shown?
Response 6: Thankyou for highlighting this. We have changed the stated magnification to 400x fold in the figure legends for each of the proteins (Figures 2-4).
Comments 7: line 352: "mod" should be explained as moderate
Response 7: Thankyou, as you say this will be easier for the reader therefore where ‘mod’ was written we have changed it to moderate (see lines 403, 458, 481).
Comments 8: Discussion: The discussion section explains the targets investigated very well. The limitations of the study could be presented. Could tissues (bone?) be included into the analysis with basal expression of the respective DEG products? Were the different locations (axial/appendicular) compared? There is distinguished between nuclear and cytoplasmic expression of the targets but the interpretation is not clear.
Response 8: We have addressed limitations of the study throughout the paper however we have now added an additional limitations and future directions section to the discussion (lines 804-839).
Yes, comparisons between axial/appendicular locations for the nuclear and cytoplasmic H-scores have been conducted in this study, however there were no significant findings from this analysis, the results are presented though (see supplementary figure A1, line 899-913).
We have now added a more detailed section on H-scoring and nuclear/cytoplasmic staining as we recognise that the less informative ‘overall H-score’ is more commonly used. The cytoplasmic vs nuclear were manually scored and double scored which is time intensive but necessary to get the more detailed results required for this deeper analysis (lines 158-174, 276-278).
Comments 9: Minor: blanks are lacking or surplus (e.g., lines 311 versus 313 etc., 351), line 455 versus 463 connexin should either start with capital or normal sized letter. The abbreviation OSA is not consistently used.
Response 9: Thankyou for highlighting this. All ‘osteosarcoma’ has been changed to the abbreviated OSA throughout. The KEGG links have been altered to ensure there are no blanks (all within section 3.1.1). Connexin43 has been changed to its gene alias, GJA1 and capitalisation has been altered where deemed necessary. All other genes/protein names have been checked to ensure they are in the correct format (italicised for gene names and not when talking about the protein).
Round 2
Reviewer 1 Report
Comments and Suggestions for Authors
Thank you for considering the comments raised during the first reviewing, however modifications made by the authors are very much limited to text editing and my may concern remains. Indeed, the title mention a comparative study between dogs and humans regarding OS. This comparison is only limited to a list of genes that are very limited and not functionnaly tested. It would have been essential to at least compare the selected list (dogs vs humans) by IHC.
Author Response
Reviewer: Thank you for considering the comments raised during the first reviewing, however modifications made by the authors are very much limited to text editing and my may concern remains. Indeed, the title mention a comparative study between dogs and humans regarding OS. This comparison is only limited to a list of genes that are very limited and not functionnaly tested. It would have been essential to at least compare the selected list (dogs vs humans) by IHC.
Authors: Thank you for approving of the first round of comments. We agree, the bulk of the study concentrates on the human versus canine element via RNASeq analysis and the discussion on comparisons and implications, but the clinically relevant protein analysis is in canine tissue as we have access to those samples. Sadly, the entire list is very large and represents a very large grant given the antibodies, tissues, consumables and time required and we hope to achieve more grants to study these in the future.
Reviewer 2 Report
Comments and Suggestions for Authors
The manuscript can be accepted, and the authors have addressed all my comments.
Author Response
Reviewer: The manuscript can be accepted, and the authors have addressed all my comments.
Authors: Thank you for much.